# Effect of guided Ahmed glaucoma valve implantation on corneal endothelial cells: A 2-year comparative study

**Ji Hyoung Chey[1], Chang Kyu Lee[1,2]***

**1** Department of Ophthalmology, Ulsan University Hospital, University of Ulsan College of Medicine, Ulsan, South Korea, **2** Biomedical Research Center, Ulsan University Hospital, University of Ulsan College of Medicine, Ulsan, South Korea

\* coolleo@uuh.ulsan.kr

## Abstract

### Purpose

To compare the effects of guided and non-guided Ahmed glaucoma valve (AGV) implantation on the corneal endothelium.

### Methods

Medical records of patients who underwent AGV implantation in the anterior chamber (AC) were reviewed retrospectively. The eyes were divided into two groups depending on the use of a guidance technique with spatula and a 4–0 nylon intraluminal stent. Specular microscopy was performed to measure corneal endothelial cell density (ECD) loss after surgery, and the rate of ECD change was calculated. Tube parameters were measured using anterior segment optical coherence tomography (AS-OCT).

### Results

The ECD loss during 2 years of follow-up was significantly lower in the guided AGV implantation (gAGV) group than in the non-guided implantation (ngAGV) group, and the rate of ECD change was -0.62 ± 1.23 and -1.42 ± 1.57%/month in the gAGV and ngAGV groups, respectively (p = 0.003). The mean tube–cornea distance (TCD) and mean tube–cornea angle (TCA) were significantly greater in the gAGV group than in the ngAGV group. The frequency of tube repositioning within 2 years after surgery was 0% in the gAGV group and 12.66% in the ngAGV group (p = 0.005).

### Conclusions

The use of a guidance technique can reduce corneal endothelial loss during the first 2 years after AGV implantation in the AC. The tube was positioned at a more distant and wider angle

**Data Availability Statement:** All relevant data are within the paper.

**Funding:** The authors received no specific funding for this work.

**Competing interests:** The authors have declared that no competing interests exist.

from the cornea in the eyes of the gAGV group, which may have contributed to the reduced need for tube repositioning to prevent corneal decompensation.

## Introduction

The frequency of glaucoma drainage device (GDD) surgery in cases of medically uncontrolled glaucoma is increasing worldwide. According to the 2012 Tube Versus trabeculectomy (TVT) study, in comparison with trabeculectomy with mitomycin C, tube shunt surgery showed comparable intraocular pressure (IOP) reduction and requirement for additional glaucoma medications, as well as a higher success rate over 5 years of follow-up in eyes with previous ocular surgery [1].

Although the effectiveness and safety of GDD surgery have been proven for decades, corneal decompensation is one of the most common long-term complications after GDD surgery, and it has devastating consequences. Variable rates of corneal decompensation have been reported in previous studies [2–4]. The TVT study showed that 16% of tube patients developed persistent corneal edema with 8% undergoing subsequent keratoplasty over 5 years of follow-up [5]. These rates were twice as high as those in the trabeculectomy group, which implies that GDD surgery possesses a higher risk of corneal decompensation than trabeculectomy. The Ahmed Baerveldt Comparison study also found the corneal decompensation rate to be 20% during 5 years of follow-up, although 11% of them were likely attributable to implant other than pre-existing corneal pathology [6]. The cumulative probability of corneal decompensation after Ahmed glaucoma valve (AGV) implantation was 3.3% over 5 years in the study by Kim et al. [7], but the difference was not significant in comparison with control eyes after 2 years. However, a recent study by Beatson et al. reported that the cumulative probability of developing decompensation after GDD surgery at 3, 6, 9 years was 4.7%, 9.2%, and 14.8%, respectively, with the risk persisting over time [8].

Recent studies on tube parameters found that a longer tube–cornea distance (TCD) and deeper tube–cornea angle (TCA) induced less endothelial cell density (ECD) loss after AGV implantation in the anterior chamber (AC), emphasizing the importance of accurately positioning the tube [9, 10]. However, to the best of our knowledge, no comparative studies have been conducted on the surgical methods that may be useful for tube positioning in the AC.

In this study, we introduced a guidance technique using a spatula and a 4–0 nylon intraluminal stent during tube insertion in the AC. This method enables direct identification of the lowest entry point in the angle and firmly supports the flexible tube from kinking and bending when the tube enters the AC. We compared the corneal ECD change, surgical outcomes, and frequency of tube repositioning between the guided AGV implantation (gAGV) and non-guided AGV implantation (ngAGV) groups during 2 years of follow-up to evaluate the effect of guided AGV implantation on the corneal endothelium.

## Methods

This retrospective study was approved by the Institutional Review Board of our university hospital (IRB number: 2022-05-030-001) and was conducted in accordance with the tenets of the Declaration of Helsinki. The requirement for written informed consent was waived owing to the retrospective nature of the study.

We reviewed the medical records of patients with glaucoma who were unresponsive to maximal medical treatment or had intolerable complications of anti-glaucoma drugs who

underwent AGV implantation between 2016 and 2021. Patients with at least six months of postoperative follow-up were included in this study. Exclusion criteria included previous corneal transplant, preexisting corneal diseases that could affect the corneal endothelium, preoperative ECD less than 1200 cells/mm$^2$, previous AGV implantation in the same eye, previous intraocular surgery within 6 months, and previous or concurrent complicated cataract surgery. Patients who were unable to complete the examinations required for this study were also excluded. If patients underwent other intraocular surgeries during follow-up, data before surgery were collected.

Patient demographics and ocular characteristics were collected by reviewing medical records, including age, sex, glaucoma diagnosis, laterality, number of glaucoma medications, best-corrected visual acuity (BCVA), IOP, systemic diseases, previous intraocular surgery, central corneal thickness, axial length, and AC depth. Postoperative data included BCVA, IOP, anterior segment inflammation, the number of glaucoma medications, and surgical complications. In both groups, surgical times (in seconds) were measured from the start of using a spatula in the guided group and the start of using a 23-gauge needle in the non-guided group; surgical time measurements in both groups were stopped at the time of successful AC tube insertion.

Corneal endothelial evaluation was performed using a non-contact specular microscope (Topcon SP-3000P; Topcon Corp., Tokyo, Japan) in patients within 1 month before AGV implantation and repeated postoperatively until the last day of follow-up. The central corneal area was examined three times consecutively to obtain the average measurements, while the patients fixed their gaze at the target in the instrument. The manual center-dot method was used to measure the central corneal ECD, coefficient of variation (CV), and hexagonality, marking at least 50 contiguous endothelial cells [11].

Anterior segment optical coherence tomography (AS-OCT) (Triton SS-OCT; Topcon Corp., Tokyo, Japan) was performed 3–6 months after surgery to determine tube location in the AC. Patients were asked to fixate on an internal fixation target. Images that lacked artifacts of eye motion and blinking were included in the analyses, and the best-quality images were obtained using methods similar to those described by Hau et al. [12]. The scanning axis was positioned along the tube such that the spatial relationship between the tube and other structures of the AC could be clearly visualized. The perpendicular TCD between the anterior tip of the tube and the cornea, the length of the tube (TL) from the point of entrance into the AC to the anterior tip of the tube, and the TCA between the posterior corneal surface and anterior surface of the tube were measured using a built-in ruler and angle indicator.

Repositioning of the AGV tube was performed if the postoperative ECD decreased by more than 50% of the preoperative ECD and reached less than 1000 cells/mm$^2$. The previous tube was withdrawn from the AC and inserted into the ciliary sulcus (CS) along the newly created sclerotomy path by using a 23-gauge needle.

All surgeries were performed by a single surgeon (CKL), and all AGVs were FP-7. The AGV tube was placed in the AC in all cases by using a previously reported standard method [13]. The method was gradually changed from non-guided to guided AGV implantation since 2020 and two surgical methods were not used together during the same period.

Under sub-Tenon anesthesia, a fornix-based flap of the conjunctiva and Tenon's capsule was created in the superior temporal or nasal quadrants. Two partial-thickness scleral flaps were made with approximately one-third to half of the total thickness of the sclera to stabilize the tube (Fig 1A). The tube was flushed with a balanced salt solution to check patency. Partial ligation of the tube and 4–0 nylon intraluminal stent was performed at the proximal site using 8–0 vicyl, and the stent was withdrawn. The AGV was anchored between the superior rectus and lateral or medial rectus muscles using 7–0 nylon, with the anterior edge of the plate at least

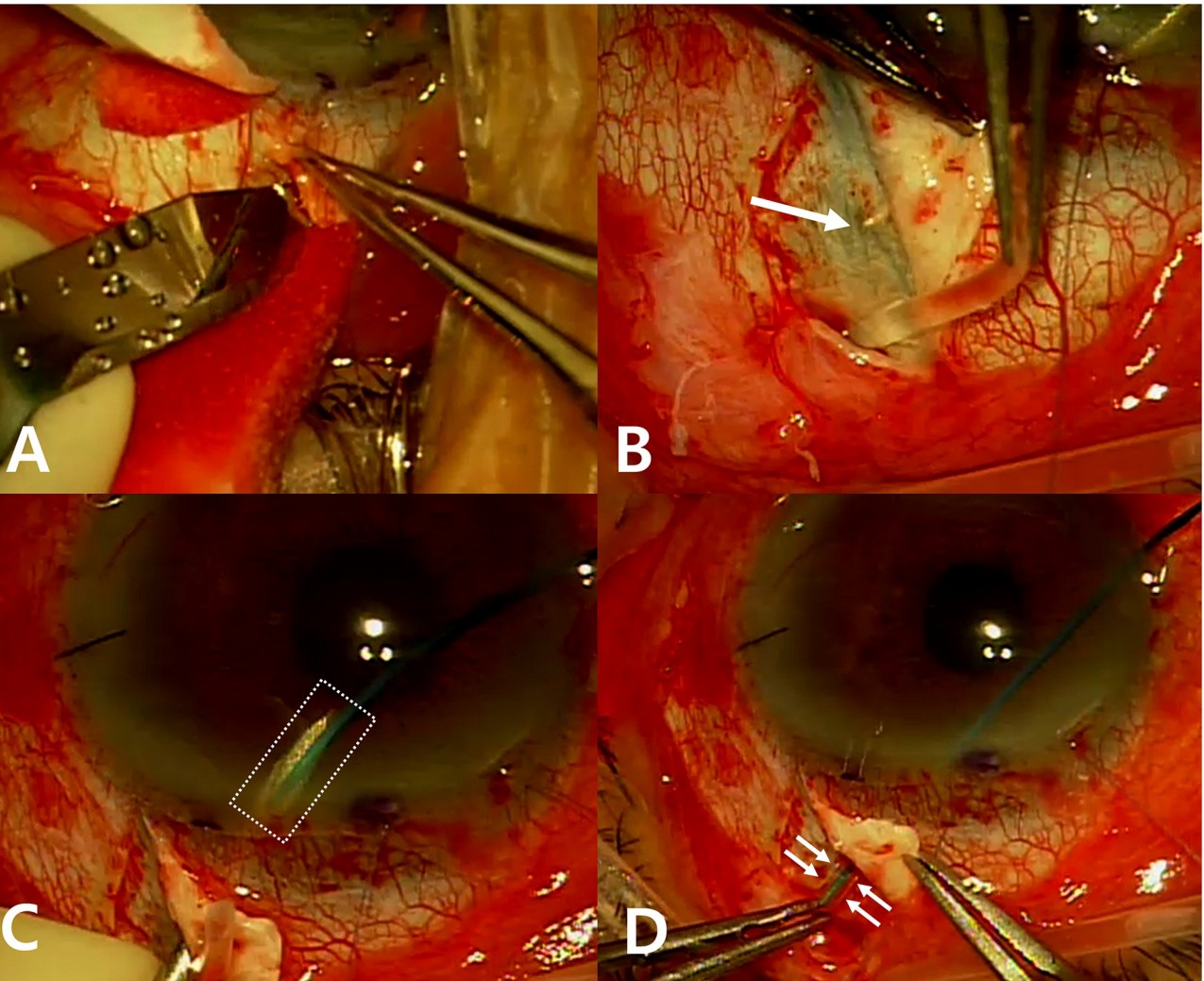

**Fig 1. Intraoperative photographs showing the main steps in the process of guided tube insertion of Ahmed glaucoma valve (AGV).** (A) Partial-thickness scleral flaps are made by scleral incision to prevent the tube from being exposed or out of position. (B) The spatula tip is gently rubbed to mark the insertion site under the scleral flap. The rubbed sclera becomes thin and transparent, revealing the spatula tip underneath (a white arrow). (C) Pre-cut 4–0 nylon with a diagonal cut end is docked into the 23G needle in the AC (a dashed white rectangle). (D) Once the 4–0 nylon is outside the eyeball, it is placed into the trimmed AGV tube, and the tube is gently inserted to the AC together with intraluminal nylon. Small white arrows indicate the tube boundary just before entering the sclerotomy site, and the 4–0 nylon stent is inside the tube.

8 mm posterior to the limbus. The tube was trimmed to an adequate length by using a bevel-up edge. In the ngAGV group, a tube was inserted into the AC along the scleral tunnel created 2–3 mm posterior to the limbus under the scleral flap by using a bent 23-gauge needle, followed by an ophthalmic viscoelastic device (OVD) injection. In the gAGV group, a spatula with a round tip was inserted through the nasal paracentesis site after the OVD injection, and the spatula tip was gently rubbed to mark the insertion site for the tube under the scleral flap to be visible from the outside of the eyeball (Fig 1B). Notably, placing the spatula too low would capture the peripheral iris and show how it was dragged. A bent 23-gauge needle was inserted into the AC from the marking location and pre-cut 4–0 nylon was docked into the needle inside the AC (Fig 1C). After 4–0 nylon was pulled outside the insertion site, it was placed into the trimmed tube, and the tube was gently inserted into the AC by using nylon as a

guide (Fig 1D). In both groups, the tube was covered with a scleral flap sutured using 10–0 nylon, and the conjunctiva and Tenon's capsule were closed using 8–0 vicryl. Topical steroids and antibiotic eye drops were administered for 8 weeks after surgery.

All data were analyzed using SPSS version 24.0 (SPSS, Inc., Chicago, IL, USA). Demographic data and surgical results were compared between the two groups by using Student's t-test for continuous variables and Pearson's chi-square test or Fisher's exact test for categorical variables. Tube parameters and postoperative corneal ECD measurements were compared between the two groups by using Student's t-test. Paired t-tests were used to compare pre- and postoperative ECD. The rate of ECD change after AGV implantation was determined using linear regression, and the slope of each regression equation represented the rate of ECD change in %/month. The rates of ECD change between the two groups were compared using Student's t-test. Statistical significance was set at $P < 0.05$.

## Results

A total of 135 eyes from 135 patients were included in this study, with 56 and 79 eyes in the gAGV and ngAGV groups, respectively. Preoperative characteristics of the patients showed no significant difference between the two groups, including age, sex, laterality, BCVA, IOP, axial length, AC depth, central corneal thickness, number of glaucoma medications, glaucoma diagnosis, systemic diseases, previous intraocular surgery and total follow-up period ($p > 0.05$, Table 1). This study included combined cataract surgery and AGV implantation, i.e., triple surgery, and the percentage of triple surgery in the gAGV and ngAGV groups was 41.07% and 26.58%, respectively, with no significant intergroup difference ($p = 0.077$, Table 1).

The surgical results in both groups showed comparable postoperative BCVA, IOP at different time intervals, number of glaucoma medications, and number of topical carbonic anhydrase inhibitors (CAI) ($p > 0.05$, Table 2). Postoperative AC cells showed no significant difference between the two groups, and iridocorneal touch did not occur in either group. Hypotony and hyphema were the two most common complications, accounting for one-quarter of the postoperative complications. The percentages of these complications were not significantly different between the two groups. Other postoperative complications also showed comparable results between the two groups, except for flat AC, which required viscoelastic injection. The mean surgical time was higher in the gAGV group, but the difference was not statistically significant (Table 2). Repositioning of the AGV tube in the CS was required in 0 eyes in the gAGV group and in 10 eyes (12.66%) in the ngAGV group, showing a significant difference between the groups ($p = 0.005$, Table 2).

Comparisons of postoperative corneal ECD showed statistically significant differences between the gAGV and ngAGV groups (Table 3). While the two groups showed no significant differences in preoperative corneal endothelial cell measurements, including ECD, CV, and hexagonality, the postoperative ECD at the final visit was 1949.39 ± 488.95 cells/mm$^2$ in the gAGV group which was significantly higher than the corresponding value in the ngAGV group (1571.95 ± 708.82 cells/mm$^2$; $p = 0.001$, Table 3). The percentage of postoperative ECD loss was 7.91 ± 11.72% in the gAGV group and 29.34 ± 22.20% in the ngAGV group ($p < 0.001$, Table 3). The rate of ECD change per month was significantly lower in the gAGV group than in the ngAGV group (-0.62 ± 1.23% and -1.42 ± 1.57%, respectively, $p = 0.003$, Table 3).

Fig 2 shows the mean percentage changes of residual corneal ECD over time after AGV implantation in the two groups. The loss of ECD showed linear patterns overall in both groups,

**Table 1. Baseline characteristics of the patients.**

|  | Total | gAGV | ngAGV | p value* |
|---|---|---|---|---|
|  | n = 135 | n = 56 | n = 79 |  |
| Age (years) | 69.3 ± 10.9 | 70.1 ± 10.3 | 68.8 ± 11.3 | 0.493 |
| Male / Female (n) | 90 / 45 | 37 / 19 | 53 / 26 | 0.888† |
| Right / Left (n) | 70 / 65 | 30 / 26 | 40 / 39 | 0.740† |
| Lens status [n (%)] |  |  |  | 0.718† |
| Phakia | 8 (5.93) | 4 (7.14) | 4 (5.06) |  |
| Pseudophakia | 127 (94.1) | 52 (92.9) | 75 (94.9) |  |
| Triple[a] [n (%)] | 44 (32.6) | 23 (41.1) | 21 (26.6) | 0.077† |
| Follow-up period (months) | 17.3 ± 5.99 | 15.8 ± 6.49 | 18.4 ± 5.41 | 0.100 |
| Preoperative BCVA (LogMAR) | 0.96 ± 0.90 | 0.84 ± 0.88 | 1.05 ± 0.90 | 0.181 |
| Preoperative IOP (mmHg) | 27.3 ± 8.77 | 26.4 ± 7.95 | 28.0 ± 9.30 | 0.322 |
| Preoperative number of glaucoma medication (n) | 3.57 ± 0.83 | 3.64 ± 0.82 | 3.52 ± 0.85 | 0.397 |
| AL (mm) | 23.8 ± 1.21 | 23.9 ± 1.01 | 23.8 ± 1.34 | 0.836 |
| ACD (mm) | 3.52 ± 0.78 | 3.49 ± 0.91 | 3.55 ± 0.68 | 0.663 |
| CCT (μm) | 534.0 ± 38.2 | 538.0 ± 34.8 | 531.2 ± 40.4 | 0.313 |
| Glaucoma diagnosis [n (%)] |  |  |  | 0.346‡ |
| POAG | 39 (28.9) | 19 (33.9) | 20 (25.3) |  |
| NVG | 36 (26.7) | 12 (21.4) | 24 (30.4) |  |
| Pseudoexfoliation | 26 (19.3) | 13 (23.2) | 13 (16.5) |  |
| Uveitis | 8 (5.93) | 1 (1.79) | 7 (8.86) |  |
| PACG | 8 (5.93) | 4 (7.14) | 4 (5.06) |  |
| Others[b] | 18 (13.3) | 7 (12.5) | 11 (13.9) |  |
| Systemic diseases [n (%)] |  |  |  | 0.080‡ |
| HTN | 62 (45.9) | 37 (66.1) | 25 (31.7) |  |
| DM | 44 (32.6) | 17 (30.4) | 27 (34.2) |  |
| Autoimmune | 6 (4.44) | 2 (3.57) | 4 (5.06) |  |
| Previous intraocular surgery [n (%)] |  |  |  | 0.842† |
| Phaco /c PCL | 83 (65.9) | 29 (65.9) | 54 (64.3) |  |
| TLE | 13 (10.2) | 4 (9.09) | 9 (10.7) |  |
| PPV | 17 (13.3) | 7 (15.9) | 10 (11.9) |  |
| Others[c] | 15 (11.7) | 4 (9.09) | 11 (13.1) |  |

* = Results of Student's t-test.

† = Results of Pearson's chi-square test.

‡ = Results of Fisher's exact test.

Continuous variables are presented as means with standardized deviations. Categorical variables are presented as frequencies and percentages.

a = Combined phacoemulsification with IOL implantation and Ahmed glaucoma valve implantation.

b = Other glaucoma diagnoses include angle recession glaucoma, steroid-induced glaucoma, and mixed-mechanism glaucoma.

c = Other intraocular surgeries including bleb needling, intravitreal injection, and laser iridotomy.

gAGV, guided Ahmed glaucoma valve implantation; ngAGV, non-guided Ahmed glaucoma valve implantation;

BCVA, best-corrected visual acuity; IOP, intraocular pressure; POAG, primary open angle glaucoma; NVG, neovascular glaucoma; PACG, primary angle closure glaucoma; HTN. hypertension; DM, diabetes mellitus; AL, axial length; ACD, anterior chamber depth; CCT, central corneal thickness; Phaco /c PCL, phacoemulsification with posterior chamber IOL implantation; TLE, trabeculectomy; PPV, pars plana vitrectomy.

with a regression coefficient (B) of -0.399% in the gAGV group and -1.149% in the ngAGV group (p = 0.009 and < 0.001, respectively).

The tube parameters measured by AS-OCT are listed in Table 4. Both groups had comparable TL, while the gAGV group had a longer TCD (1101.32 ± 376.99 mm) and greater TCA

**Table 2. Comparison of surgical results after guided and non-guided Ahmed glaucoma valve implantation.**

|  | Total | gAGV | ngAGV | *p* value* |
|---|---|---|---|---|
|  | **n = 135** | **n = 56** | **n = 79** |  |
| Postoperative BCVA (LogMAR) | 0.97 ± 0.92 | 0.83 ± 0.88 | 1.08 ± 0.95 | 0.118 |
| Postoperative IOP (mmHg) |  |  |  |  |
| 12 months | 12.4 ± 3.74 | 12.9 ± 4.36 | 12.2 ± 3.45 | 0.386 |
| 24 months | 13.7 ± 4.47 | 14.8 ± 4.97 | 13.4 ± 4.37 | 0.355 |
| Final visit | 13.5 ± 4.45 | 14.1 ± 4.21 | 13.1 ± 4.61 | 0.228 |
| Postoperative number of glaucoma medication (n) | 1.56 ± 1.06 | 1.54 ± 1.04 | 1.58 ± 1.07 | 0.802 |
| Postoperative number of topical CAI (n) | 0.45 ± 0.50 | 0.52 ± 0.50 | 0.41 ± 0.49 | 0.197 |
| Postoperative AC cell[a] |  |  |  |  |
| At 1 day | 1.01 ± 0.79 | 1.05 ± 0.70 | 0.97 ± 0.85 | 0.568 |
| At 1 week | 0.56 ± 0.87 | 0.45 ± 0.74 | 0.65 ± 0.95 | 0.191 |
| At 1 month | 0.04 ± 0.21 | 0.05 ± 0.23 | 0.04 ± 0.19 | 0.668 |
| Postoperative complications ≤ 1 month [n (%)] |  |  |  | 0.642‡ |
| Hypotony[b] | 16 (11.9) | 6 (4.44) | 10 (12.7) | 0.729† |
| Hyphema | 19 (14.1) | 6 (4.44) | 13 (16.5) | 0.354† |
| Flat AC[c] | 7 (5.19) | 0 | 7 (8.86) | **0.021‡** |
| CD | 5 (3.70) | 2 (1.48) | 3 (3.80) | 0.659‡ |
| IOL related | 1 (0.74) | 0 | 1 (1.27) | 0.585‡ |
| Vitreous hemorrhage | 4 (2.96) | 1 (1.79) | 3 (3.80) | 0.448‡ |
| Vitreous prolapse | 4 (2.96) | 1 (1.79) | 3 (3.80) | 0.448‡ |
| Iris related | 2 (1.48) | 1 (1.79) | 1 (1.27) | 0.659‡ |
| Iridocorneal touch (n) | 0 | 0 | 0 |  |
| Surgical time (s) | 251.7 ± 192.1 | 267.7 ± 168.4 | 240.0 ± 208.2 | 0.428 |
| Repositioning of tube [n (%)] | 10 (7.41) | 0 (0) | 10 (12.7) | **0.005‡** |

* = Results of Student's t-test.

† = Results of Pearson's chi-square test.

‡ = Results of Fisher's exact test.

Continuous variables are presented as means with standardized deviations. Categorical variables are presented as frequencies and percentages.

a = measurement by the standardization of uveitis nomenclature (SUN) grading system.

b = intraocular pressure ≤ 5 mmHg by Goldmann applanation tonometry.

c = viscoelastic injection into the AC required.

gAGV, guided Ahmed glaucoma valve implantation; ngAGV, non-guided Ahmed glaucoma valve implantation;

BCVA, best-corrected visual acuity; IOP, intraocular pressure; CAI, carbonic anhydrase inhibitor; AC, anterior chamber; CD, choroidal detachment; IOL, intraocular lens.

(32.33 ± 6.65˚), than the ngAGV group (863.35 ± 357.31 mm and 28.51 ± 5.91˚, respectively, p = 0.004 and p = 0.007, respectively). Representative cases from the gAGV and ngAGV groups showed different tube positioning for each method (Fig 3).

## Discussion

In this retrospective study, we compared and analyzed the effect of AGV implantation with and without guided tube insertion into the AC on corneal ECD and surgical outcomes. Previous studies have reported corneal ECD loss after AGV implantation in the AC [14, 15]. Kim et al. [14] reported a 10.5% reduction in the central ECD after 12 months, and Lee et al. [15] reported a 15.3% and 18.6% reduction in average ECD at 12 and 24 months after surgery, respectively.

**Table 3. Comparisons of preoperative and postoperative corneal endothelial cell measurements between the guided and non-guided Ahmed glaucoma valve implantation groups.**

| | Total | gAGV | ngAGV | $p$ value[*] |
|---|---|---|---|---|
| | **n = 135** | **n = 56** | **n = 79** | |
| Preoperative ECD (cells/mm$^2$) | 2140.6 ± 509.6 | 2111.0 ± 473.8 | 2161.6 ± 535.8 | 0.572 |
| Preoperative CV | 29.2 ± 10.3 | 27.5 ± 10.2 | 30.4 ± 10.2 | 0.115 |
| Preoperative hexagonality (%) | 31.4 ± 41.7 | 25.9 ± 39.9 | 35.2 ± 42.7 | 0.208 |
| Postoperative ECD (cells/mm$^2$) | 1728.5 ± 652.3 | 1949.4 ± 489.0 | 1572.0 ± 708.8 | **0.001** |
| | **<0.001**[§] | **<0.001**[§] | **<0.001**[§] | |
| Postoperative ECD loss[a] (%) | 20.5 ± 21.4 | 7.91 ± 11.7 | 29.3 ± 22.2 | **<0.001** |
| Rate of monthly ECD change[b] (%) | -1.08 ± 1.49 | -0.62 ± 1.23 | -1.42 ± 1.57 | **0.003** |

[*] = Results of Student's t-test.

[§] = Results of paired t-test.

Values are presented as mean ± standard deviation.

[a] ECD loss (%) = (preoperative ECD—postoperative ECD)/preoperative ECD × 100.

[b] The rate of monthly ECD change (%) was calculated and averaged from the slope of each linear regression equation.

gAGV, guided Ahmed glaucoma valve implantation; ngAGV, non-guided Ahmed glaucoma valve implantation;

ECD, endothelial cell density; CV, coefficient of variation.

Various hypotheses have been proposed to explain the mechanism leading to corneal ECD reduction after tube shunt surgery. Mechanical factors include the foreign body effect of the silicone tube, progressive tube migration, peripheral anterior synechiae (PAS), and transient tube-corneal/uveal contact with blinking and rubbing [12, 14, 16–18]. Jets of aqueous humor fluid through the tube occurring in sync with heartbeats could result in ECD loss near the tube [19]. Chronic inflammation caused by the silicone tube and chronic trauma results in increased endothelial cell permeability and depletion of nutrients and oxygen, leading to corneal edema [19, 20]. Moreover, the aqueous humor protein concentration was increased 10-fold in a previous study, suggesting an impairment in the blood-aqueous barrier to allow

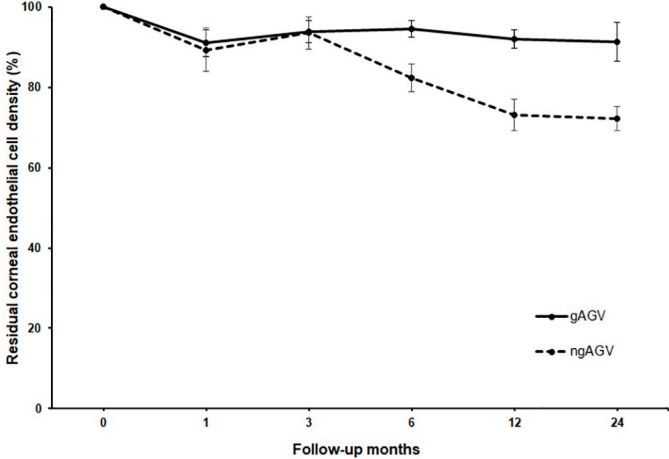

**Fig 2. Changes in residual corneal endothelial cell density (ECD) over time after Ahmed glaucoma valve (AGV) implantation in the guided AGV and non-guided AGV groups.** Dots represent mean percentages of remaining ECD in comparison with the preoperative ECD starting at 100%. Standard error bars are given at 1, 3, 6, 12, and 24 months.

**Table 4. Comparison of tube parameters between the guided and non-guided Ahmed glaucoma valve implantation groups.**

| | Total | gAGV | ngAGV | p value* |
|---|---|---|---|---|
| | n = 135 | n = 56 | n = 79 | |
| TL (mm) | 1656.5 ± 533.8 | 1702.8 ± 494.3 | 1603.4 ± 577.7 | 0.392 |
| TCD (mm) | 988.0 ± 384.6 | 1101.3 ± 377.0 | 863.4 ± 357.3 | **0.004** |
| TCA (°) | 30.5 ± 6.57 | 32.3 ± 6.65 | 28.5 ± 5.91 | **0.007** |

* = Results of Student's t-test.

Values are presented as mean ± standard deviation.

gAGV, guided Ahmed glaucoma valve implantation; ngAGV, non-guided Ahmed glaucoma valve implantation;

TL, tube length; TCD, tube–cornea distance; TCA, tube–cornea angle.

oxidative, apoptotic, and inflammatory proteins to enter the AC and alter the aqueous humor environment [21, 22].

In addition to tube-related factors, other factors can influence corneal ECD, including age, toxicity from glaucoma medication, high preoperative IOP, longer duration of high IOP before surgery, higher number of previous intraocular surgeries, and history of uveitis [23].

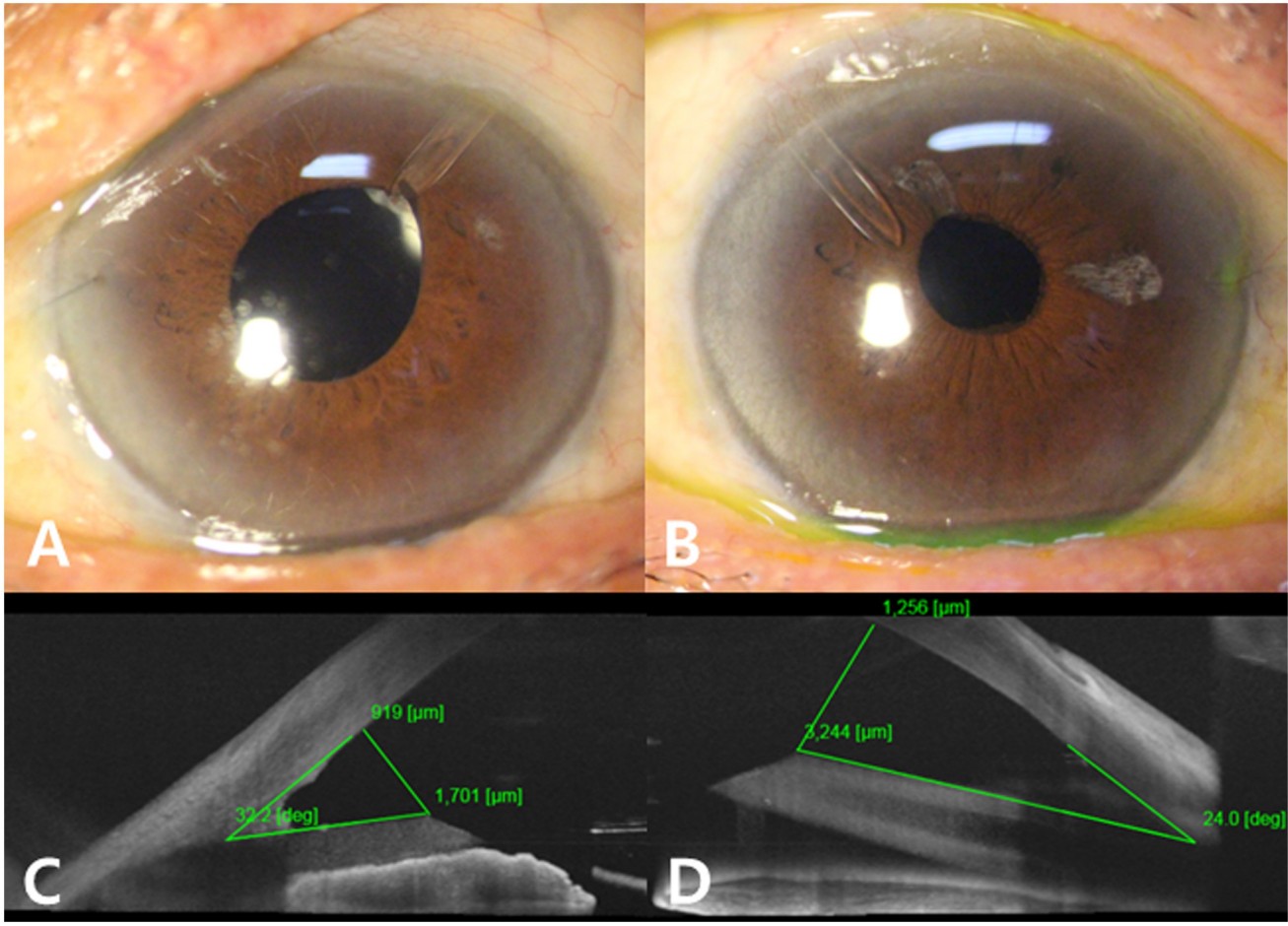

**Fig 3. Representative cases from the gAGV and ngAGV groups.** Slit-lamp photographs and AS-OCT imaging of two representative cases from the gAGV (A, C) and ngAGV (B, D) groups. AS-OCT shows the tube on the left (C) with a greater TCA and TCD than the tube on the right (D).

Differences in ECD reduction have been reported according to the type of glaucoma, wherein eyes with angle closure glaucoma and pseudoexfoliation glaucoma showed significantly decreased ECD [24, 25].

Efforts to minimize the reduction in corneal ECD after tube shunt surgery are ongoing. AGV tube insertion into the CS using the iris as a mechanical barrier has significantly lower rates of ECD change than tube insertion in the AC in previous studies [26–28]. However, tube implantation in the CS may be related to a higher incidence of postoperative hemorrhage, since the CS shows greater vascularization than the AC [29], as well as intraocular lens (IOL)-related complications and capsular bag stability and zonular weakness should also be considered, and the structural difficulty of tube insertion in the CS may be present in Asian eyes with relatively small axial lengths. Placing the tube on the par plana can minimize the effect of the tube on the corneal endothelium but is limited by the prerequisite that previous vitrectomy must be performed.

In our study, no significant differences were observed in the demographic data between the gAGV and ngAGV groups, indicating that the influence of nonsurgical factors such as age, preoperative IOP, lens status, or glaucoma types that affect corneal ECD loss can be excluded. Comparisons of surgical results also showed no significant differences in postoperative IOP and the number of glaucoma medications, including topical CAIs between the two groups, thereby excluding the influence of these factors on the corneal endothelium. We also assessed postoperative inflammation by grading AC cells up to 1 month and found that postoperative inflammation in the AC did not increase in the gAGV group in comparison with that in the ngAGV group.

Postoperative complications and surgical times were compared to ensure the safety and efficiency, respectively, of the guidance technique. Surgical time was measured in seconds only for the part where the surgical technique of the two groups was different; a spatula and a guiding stent were used in the gAGV group while the tube was directly inserted into the ngAGV group. Guided AGV implantation took an average of 30 s longer than simple tube insertion because the former required more steps; however, based on the standard deviation, the guided procedure could be completed without much deviation from the mean time. The surgical time was relatively short in many eyes in the ngAGV group, but the procedure took much longer in some eyes in which the tube was not inserted into the desired position, or the tube was bent and did not enter the AC well. Postoperative complications were not significantly different between the two groups except for flat AC, which might be caused by leakage of the sclerotomy site after multiple punctures to obtain the desired tube position in the ngAGV group. Therefore, guided AGV implantation can be considered a relatively safe surgical technique that requires little additional time and sometimes saves time by facilitating tube insertion.

A guidance technique was proposed with the goal of ideal positioning of the AGV tube in the AC as far as possible from the cornea and parallel to the iris. To assess the tube position, the tube parameters were measured and compared between the two groups. The TCD and TCA are both considered important tube parameters, but the TCD is not a fully independent variable since it is affected by the TL and TCA [9]. In this study, the mean TLs in both groups were not significantly different; therefore, the influence of TL on TCD could be excluded.

Postoperative ECDs at the final visit were significantly lower than the corresponding preoperative value in eyes of the gAGV and ngAGV groups ($p < 0.001$). However, the percentage of postoperative ECD loss was lower in the gAGV group, and the rate of ECD change was also lower in the gAGV group, which implied that less corneal endothelial damage occurred in the gAGV group within 2 years after surgery. The rate of ECD change with guided AGV implantation in our study appeared to be comparable to the rate of ECD change with tube insertion in the CS in previous studies (-0.36%/month in Kim et al. [27] and -0.72%/month in Zhang et al.

[28]). However, the overall degrees and rates of ECD change in this study were relatively higher than those reported previously. One possible explanation is that approximately 30% of the eyes included in our study had concurrent phacoemulsification, although lens status was not associated with treatment failure in the TVT study [1].

Changes in residual corneal ECD over time after AGV implantation showed a clear difference in the slope between the two groups, and the gap in the remaining ECD widened as the follow-up period increased. The transient rapid decline in ECD at 1 month and recovery at 3 months suggest that lost endothelium due to corneal injuries during surgery might be recovered by stem cells from a niche at the posterior limbus [30]. The mean percentage loss in corneal ECD was more prominent during the first 12 months than after 12 months. In fact, the mean follow-up period until tube repositioning was 13.40 ± 6.15 months in the ngAGV group, indicating the importance of careful monitoring of the remaining ECD in the first 12 months after surgery. Eyes that underwent tube repositioning within 2 years in this study had a mean ECD loss of 54.42% and a mean rate of ECD change of 4.80%/month. These figures were similar to those in the eyes included in this study that developed corneal decompensation after 2 years, with a mean ECD loss of 50.30% and a mean rate of ECD change of 4.09%/month. This result suggests that our criteria for tube repositioning are effective in selecting eyes at risk of corneal decompensation for early management.

This study had some limitations due to its retrospective nature. First, the follow-up interval and specular microscopic examination interval were not the same among the patients. Linear regression analysis was used to calculate the rate of ECD change in each eye to compensate for this limitation, and at least three ECD measurements were performed in most eyes to obtain a reliable slope. Second, the mean follow-up period in the gAGV group was shorter than that in the ngAGV group, although the difference was not significant. As the surgical method was gradually changed from simple tube insertion to a guidance-based technique, the ngAGV group had a greater chance of undergoing long-term follow-up examinations. However, we tried to compensate for this limitation by strictly obeying the inclusion criteria and limiting the data to within two years. In addition, the mean follow-up period in the gAGV group was longer than that until tube repositioning. An advantage of the gradual change in the surgical method is that there was no selection bias between the two groups because patients who visited in the specific period underwent surgery using the same surgical method regardless of their age, sex, or glaucoma diagnosis. Therefore, we excluded cases one month after the first guided AGV implantation in consideration of the learning curve and selection bias. Third, previous or concurrent cataract surgery may affect the rate of ECD change during 2 years of follow-up. However, the endothelial loss after cataract surgery is generally considered to be a one-time event [15], and we tried to minimize this limitation by excluding complicated cataract surgery and previous intraocular surgery within 6 months. Also, there was no significant difference in the number of triple surgeries between two groups although the percentage of triple surgery was higher in the gAGV group. Forth, the postoperative PAS score, which may affect the corneal endothelium was not assessed in this study. We examined the iridocorneal contact by slit-lamp examination but did not use gonioscopy. Fifth, the AGV tube may change its position and length over time in the AC; therefore, tube positioning assessment using AS-OCT may vary depending on the time of evaluation [31, 32]. In this study, AS-OCT was performed 3–6 months after surgery to access a relatively early tube position. Finally, as the mean time from GDD surgery to corneal decompensation was over 24 months in a previous study [8], longer follow-up periods would be necessary to further investigate the association with corneal decompensation.

In conclusion, in comparison with the conventional method, guided AGV implantation resulted in a lower corneal ECD loss and frequency of tip repositioning within 24 months.

Guided AGV implantation was associated with less corneal ECD loss and a lower rate of post-operative ECD change in comparison with non-guided AGV implantation. Tube parameter analysis showed that the guidance technique could be used to position the tube further from the corneal endothelium at both distances and angles. The two groups showed no difference in the frequency of postoperative complications, and tube insertion was consistently completed within the mean surgical time. Thus, guided AGV implantation is a safe and time-efficient surgical technique that may contribute to the prevention of postoperative corneal decompensation when an AGV tube is inserted into the AC.

## Supporting information

**S1 Video. Video showing the main steps in the process of guided tube insertion of Ahmed glaucoma valve.**
(WMV)

## Author Contributions

**Conceptualization:** Chang Kyu Lee.

**Methodology:** Chang Kyu Lee.

**Writing – original draft:** Ji Hyoung Chey.

**Writing – review & editing:** Chang Kyu Lee.

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
