## [Decision Letter · Decision Letter 0]

18 Oct 2022

PONE-D-22-22914Effect of Guided Ahmed Glaucoma Valve Implantation on Corneal Endothelial Cells: A 2-Year Comparative StudyPLOS ONE

Dear Dr. Lee,

Thank you for submitting your manuscript to PLOS ONE. After careful consideration, we feel that it has merit but does not fully meet PLOS ONE’s publication criteria as it currently stands. Therefore, we invite you to submit a revised version of the manuscript that addresses the points raised during the review process.

We look forward to receiving your revised manuscript.

Kind regards,

Asaf Achiron

Academic Editor

PLOS ONE

Journal Requirements:

"The authors received no specific funding for this work."

4. Thank you for stating the following in the Funding Section of your manuscript: 

"This research was supported by the Basic Science Research Program through the National Research Foundation of Korea (NRF) funded by the Ministry of Science and ICT (NRF-2017R1C1B5018279)"

We note that you have provided funding information that is not currently declared in your Funding Statement. However, funding information should not appear in the Funding section or other areas of your manuscript. We will only publish funding information present in the Funding Statement section of the online submission form. 

"The authors received no specific funding for this work."

Reviewers' comments:

Reviewer's Responses to Questions

**Comments to the Author**

1. Is the manuscript technically sound, and do the data support the conclusions?

Reviewer #1: Yes

Reviewer #2: Yes

2. Has the statistical analysis been performed appropriately and rigorously? 

Reviewer #1: Yes

Reviewer #2: Yes

3. Have the authors made all data underlying the findings in their manuscript fully available?

Reviewer #1: Yes

Reviewer #2: Yes

4. Is the manuscript presented in an intelligible fashion and written in standard English?

Reviewer #1: Yes

Reviewer #2: Yes

5. Review Comments to the Author

Reviewer #1: Chey and Lee present a study that aims to compare the effects of guided vs. non-guided Ahmed glaucoma valve (AGV) implantation.

General comment:

The main limitation of this study are its retrospective nature and lack of randomization in a prospective manner. Group sizes compared are not equal. It also seems that lens status (phakic vs pseudophakic), which may affect ECD count as well as tube location, was not taken fully into consideration. Nevertheless, it seems that there may be benefit in using the presented guided technique in terms of ECD preservation.

Additional comments:

1. Lines 54-59 – The cited 16-20% rate of corneal decompensation following tube shunt at the ABC/TVT studies seems too high. Please carefully double check the cited references.

2. Methods, Line 85: Why were eyes with previous corneal transplant or low endothelial cell count excluded? If they had a tube shunt placed, it would make sense to include them.

3. Figure 1 is unclear, especially 1B. I suggest adding arrows with better explanations and/or replacing the figure. The technique and illustrations should be presented in a clearer manner.

4. Video – I was unable to open the file.

5. Line 123 – why was a 4-0 nylon stent with 8-0 vicryl tune ligation performed when inserting AGV?

6. Methods: The main advantage of this method is marking the entry site from the AC with a spatula to allow for the bent 23G needle to enter the AC in the ideal location. I don’t think that adding the 4-0 nylon stent adds much as a guide. The marking from the AC might be challenging and even damage the angle structures if not done correctly. This should be mentioned in the discussion.

7. Results: groups are not well balanced – 56 vs. 79 eyes.

8. Results Table 1 – how many eyes were phakic vs pseudophakic at the end of procedure in each group? You only mention previous cataract surgery and it seems more individuals had combined triple surgery in the guided group. This could affect tube location as well as well as ECD count.

9. Results – all tables and figures should have figures presented with only 1 decimal point.

10. Table 2 – what is the mean follow us in each group? This could also affect ECD count.

11. Discussion – points for discussion should added as suggested above.

Reviewer #2: The authors claim that the postoperative loss of corneal endothelial cells was less in the case of guiding the tube placement during Ahmed valve implantation. Overall, I think it's an interesting study. However, since this is a retrospective study, further analysis and description would be needed. In the case of guiding a tube, it seems necessary to explain why the guiding was conducted. It should be described whether the surgical method changed by period or only in what cases. In addition, it would be better to further analyze the power of analysis according to the number of subjects.

6. PLOS authors have the option to publish the peer review history of their article (what does this mean?). If published, this will include your full peer review and any attached files.

Reviewer #1: No

Reviewer #2: No

---

## [Author Response · Author response to Decision Letter 0]

7 Nov 2022

Dear Editor & reviewers

The manuscript has been rechecked, and appropriate changes have been made in accordance with reviewers’ suggestions. Responses to their comments have been prepared and attached below.

We thank the editor and reviewers for pertinent suggestions and insights, which have enriched the manuscript and produced a better and more balanced account of our research.

We hope that the revised manuscript is now suitable for publication in your journal.

Reviewer #1: Chey and Lee present a study that aims to compare the effects of guided vs. non-guided Ahmed glaucoma valve (AGV) implantation.

General comment:

The main limitation of this study are its retrospective nature and lack of randomization in a prospective manner. Group sizes compared are not equal. It also seems that lens status (phakic vs pseudophakic), which may affect ECD count as well as tube location, was not taken fully into consideration. Nevertheless, it seems that there may be benefit in using the presented guided technique in terms of ECD preservation.

Thank you for your attentive review. 

As you pointed out, this study was conducted in a retrospective manner. We stated limitations in discussion that the follow-up and specular microscopic examination intervals were not the same among the patients. However, we did not simply subtract postECDs from preECDs to assess the ECD loss but rather calculated the rate of ECD change of each eye using linear regression to compensate this limitation. 

The main reason for the difference in the number of two groups was a gradual shift in surgical methods from simple tube insertion to a guidance-based technique. Nevertheless, the group size was sufficient in this comparative analysis, and statistical significance was well proven in Results. Previous other study comparing anterior chamber vs ciliary sulcus AGV tube placement also had different group sizes due to gradual change in surgical methods (ref. 27 in manuscript). It is believed to be the best way to randomize a single surgeon’s surgical methods since patients were assigned to guided or non-guided groups depending on the specific period they had surgery, not on their age, sex, glaucoma diagnosis or glaucoma severity. Thus, there was no selection bias between the two groups, as mentioned in Discussion line 345-347. 

Lens status was not clearly demonstrated in the original manuscript. Therefore, we added lens status in Table 1. Phakic eyes undergone a combined phacoemulsification with AGV implantation were included as ‘pseudophakia’ because the eye was pseudophakic at the time of tube insertion.

 We did not exclude phakic eyes with AGV implantation because our focus was to compare postoperative ECD changes in eyes according to surgical methods, regardless of their lens status. 

As you pointed out, lens status might affect tube position because the anterior chamber depth generally gets deepen after cataract surgery. However, preoperative anterior chamber depth was 3.49 ± 0.91 mm in the gAGV group and 3.55 ± 0.68 mm in the ngAGV group, showing no significant difference (p = 0.663, Table 1). Moreover, difference in lens status between two groups was statistically not significant (p = 0.718, Table 1).

Damage to the corneal endothelium after cataract surgery has been reported in various studies. Many authors concluded that endothelial loss was the greatest within a month after surgery, and approaches at the physiological level after three months.1,2 On the other hand, some authors claim that the rate of endothelial loss after cataract surgery was slightly higher than the physiological level after 2 years of follow-up.3 Factors known to influence the amount of EC loss after cataract surgery are any complications during cataract surgery, phaco time, and energy.4

Previous studies on ECD change after AGV implantation also included eyes with different lens status and eyes with concurrent surgery. Zhang et al. compared the effect of tube location with sulcus vs. AC and phakic eyes undergone a concurrent phacoemulsification were included. (ref. 28 in manuscript) Kim et al. and Lee et al. also included both phakic and pseudophakic eyes to evaluate changes in ECD after AGV implantation, because they claimed that damage to the corneal endothelium after cataract surgery is rather considered to be a one-time event contrast to AGV surgery that has the risk persisting for endothelial damage (ref. 7, 8, 15 in manuscript).

 In this study, we already tried to minimize the effect of cataract surgery on ECD change by excluding patients who had previously undergone intraocular surgery within 6 months and including only those who underwent uncomplicated cataract surgery. These were not clearly stated in the origianl manuscript, so we revised the sentence in Methods line 88-89:

“Exclusion criteria included previous corneal transplant, preexisting corneal diseases that could affect the corneal endothelium, preoperative ECD less than 1200 cells/mm2, previous AGV implantation in the same eye, and any previous intraocular surgery within 6 months.” �

“Exclusion criteria included previous corneal transplant, preexisting corneal diseases that could affect the corneal endothelium, preoperative ECD less than 1200 cells/mm2, previous AGV implantation in the same eye, previous intraocular surgery within 6 months, and previous or concurrent complicated cataract surgery.” 

In conclusion, lens status may affect the tube position but, in our study, phakia/pseudophakia numbers and anterior chamber depths were not significantly different between two groups, and majority of eyes were pseudophakia at the time of tube insertion.

ECD loss after cataract surgery is evident but the period of significant decline varies from study to study, although most study claimed that ECD loss was approached at physiological loss level after three months. Thus, an additional limitation existed in our study because lens status and numbers of combined cataract surgery between two groups were not the same. However, we tried to minimize this limitation by strictly obeying the exclusion criteria, and there was no significant difference in the number of triple surgeries between two groups (p = 0.077, Table 1). 

We added sentences in Discussion to include points above: line 286-287. Line 348-353.

1Oxford Cataract Treatment and Evaluation Team (OCTET). Long-term corneal endothelial cell loss after cataract surgery. Results of a randomized controlled trial. Arch Ophthalmol. 1986;104:1170-5. 

2Liesegang TJ, Bourne WM, Ilstrup DM. Short- and longterm endothelial cell loss associated with cataract extraction and intraocular lens implantation. Am J Ophthalmol. 1984;97:32-9.

3Long-term evaluation of endothelial cell loss after phacoemulsification. Lesiewska-Junk H, Kałuzny J, Malukiewicz-Wiśniewska G. Eur J Ophthalmol. 2002;12:30-3.

4Hayashi K, Hayashi H, Nakao F, Hayashi F. Risk factors for corneal endothelial injury during phacoemulsification. J Cataract Refract Surg. 1996;22:1079-84.

Additional comments:

1. Lines 54-59 – The cited 16-20% rate of corneal decompensation following tube shunt at the ABC/TVT studies seems too high. Please carefully double check the cited references.

Ref. 5 in manuscript is “Tube Versus Trabeculectomy Study Group. Postoperative complications in the Tube Versus Trabeculectomy (TVT) study during five years of follow-up.” conducted by Gedde et al. In this paper, table 3 shows late postoperative complications occurring more than 1 month after surgery. ‘Persistent corneal edema’ was shown at the first row of the table accounting for 16 % of the tube group. (table shown in the below) 

Notably, the term ‘persistent corneal edema’ is used instead of corneal decompensation, and cases were included that occurred more than 1 month after surgery, which is relatively a short period after surgery. This may explain the high percentage presented in this study.

Ref. 6 in manuscript is “Postoperative Complications in the Ahmed Baerveldt Comparison Study During Five Years of Follow-up.” conducted by Budenz et al. In this paper, table 2 shows late complications in the ABC study, and cumulative proportions of ‘corneal edema-All’ are 20.1 % and 20.4 % for each group. However, they also added ‘corneal edema – likely attributable to implant’ just below, showing cumulative proportions of 11.9 % and 11.7 % for each. (table shown in the below)

 They commented in discussion that “…We found a 20% rate of persistent corneal edema after tube implantation in the current study at 5 years. This was similar to the 5-year rate in the tube group in the TVT study, which was 16%. We did not find a difference between the two treatment arms in the ABC study, however. When we examined the reason for corneal edema, half of the cases had a reason other than the presence of a tube in the anterior chamber that could have explained the corneal edema such as pre-existing corneal transplants which could have failed, pre-existing corneal diagnoses such as ICE syndrome, or the presence of an anterior chamber intraocular lens, all of which are equally as likely to cause persistent corneal edema.” 

Therefore, we added some descriptions about decompensation rate of the ABC study in line 59-60 in Introduction.

The Ahmed Baerveldt Comparison study also found the corneal decompensation rate to be 20 % during 5 years of follow-up.[6] �

The Ahmed Baerveldt Comparison study also found the corneal decompensation rate to be 20 % during 5 years of follow-up, although 11% of them were likely attributable to implant other than pre-existing corneal pathology.[6]

To avoid confusion, we revised the sentence in line 54-55:

Variable rates of corneal decompensation (8 %-19 %) have been reported in previous studies.[2-4] �

Variable rates of corneal decompensation have been reported in previous studies.[2-4]

2. Methods, Line 85: Why were eyes with previous corneal transplant or low endothelial cell count excluded? If they had a tube shunt placed, it would make sense to include them.

It would have been valuable to compare changes in ECD according to surgical methods of vulnerable eyes. However, in this study, we excluded corneal factors that could affect ECD loss regardless of surgical methods to clearly show the results. Also, since it is widely known that ECD decreases after AGV implantation, we tried to minimize corneal endothelial damage by placing the tube in the sulcus or pars plana in vulnerable eyes. Thus, eyes with previous corneal transplant or low endothelial cell counts were excluded because the tube was not placed in the anterior chamber but in the sulcus of those eyes.

3. Figure 1 is unclear, especially 1B. I suggest adding arrows with better explanations and/or replacing the figure. The technique and illustrations should be presented in a clearer manner.

We added arrows and a rectangle in figures to clearly show how and where each step works. Please understand that the resolution of video images is not very high. We tried to capture images at the best resolution representing important steps of a guidance technique. In addition, you would understand more clearly when you see the actual technique through the attached video.

(A) Partial-thickness scleral flaps are made by scleral incision to prevent the tube from being exposed or out of position. (B) The spatula tip is gently rubbed to mark the insertion site under the scleral flap. The rubbed sclera becomes thin and transparent, revealing the spatula tip underneath (a white arrow). (C) Pre-cut 4-0 nylon with a diagonal cut end is docked into the 23G needle in the AC (a dashed white rectangle). (D) Once the 4-0 nylon is outside the eyeball, it is placed into the trimmed AGV tube, and the tube is gently inserted to the AC together with intraluminal nylon. Small white arrows indicate the tube boundary just before entering the sclerotomy site, and the 4-0 nylon stent is inside the tube.

4. Video – I was unable to open the file.

We are sorry for not making the video available for the review. We will double check the video file to make sure it works. 

5. Line 123 – why was a 4-0 nylon stent with 8-0 vicryl tune ligation performed when inserting AGV?

Partial ligation of the tube and a 4-0 nylon stent was performed with 8-0 vicryl during the preparation of the AGV, before anchoring the body to the sclera. This step is a well-known method of preventing early hypotony after AGV implantation.5 It is routinely performed by the surgeon of our study, and it is not included as a unique part of a guidance technique. 

5 Lee JJ, Park KH, Kim DM, Kim TW. Clinical outcomes of Ahmed glaucoma valve implantation using tube ligation and removable external stents. Korean J Ophthalmol. 2009;23:86-92.

6. Methods: The main advantage of this method is marking the entry site from the AC with a spatula to allow for the bent 23G needle to enter the AC in the ideal location. I don’t think that adding the 4-0 nylon stent adds much as a guide. The marking from the AC might be challenging and even damage the angle structures if not done correctly. This should be mentioned in the discussion.

The 4-0 nylon stent plays a very important role in this technique. As mentioned in Introduction, it supports the flexible tube from kinking and bending when the tube enters the AC. Even if we identify the ideal entry site in the AC and create the scleral tunnel with a bent 23-G needle, we struggle much to insert the tube along the scleral tunnel because the tube is easily bent, unlike the firm needle. Moreover, when we try to insert the tube close to the peripheral iris, the tube is easily stuck by iris increasing the risk of iridodialysis, or wrong insertion to the sulcus. Therefore, it is beneficial to use a 4-0 nylon stent to firmly support the tube from the inside to make sure the tube is inserted at the ideal position. This was also reflected in surgical time as written in Discussion line 299-301; the ngAGV group showed a greater deviation in the mean surgical time, and in most of these cases, the tube was kinked and bent and suffered a lot of time during insertion to the AC. 

It should be also noted that the prevalence of PACG is higher in Asians than others6, and ethical differences show Asians have narrower and more crowded AC structures that other races7 in previous studies. Therefore, using a guide to support the tube during insertion may have more benefits in Asians as in our study, or in patients having narrow AC structure. 

We did not think the marking step causes much damage because rubbing site is where the needle eventually punctures. However, we agree that cautions must be taken during this step, so we mentioned in line 133-136 in Methods to rub the spatula tip ‘gently’ and watch out for ‘iris dragging’. This step is completed in about 5 seconds and you would better understand how it works in the attached video. Complications such as hemorrhages or excessive pigment dispersion did not happen during the marking step in all cases. 

6Primary angle closure glaucoma: What we know and what we don't know. Sun X, Dai Y, Chen Y, Yu DY, Cringle SJ, Chen J, et al. Prog Retin Eye Res. 2017;57:26-45.

7Ethnic difference of the anterior chamber area and volume and its association with angle width. Wang D, Qi M, He M, Wu L, Lin S. Invest Ophthalmol Vis Sci. 2012;53:3139-44.

7. Results: groups are not well balanced – 56 vs. 79 eyes.

Less eyes were included in the guided AGV group because surgical methods gradually changed from non-guided to guided. As a retrospective study, we could not control the group size and selected cases based on inclusion and exclusion criteria. Nevertheless, the group size was sufficient to undergo comparative analysis and prove statistical significance in the results. Differences in baseline characteristics of two groups were not statistically significant as shown in Table 1. Previous comparative studies on AGV implantation also had different group sizes due to its retrospective nature and gradual change in surgical methods. (ref. 26, 27 in manuscript) 

8. Results Table 1 – how many eyes were phakic vs pseudophakic at the end of procedure in each group? You only mention previous cataract surgery and it seems more individuals had combined triple surgery in the guided group. This could affect tube location as well as well as ECD count.

Answers to the same question are given below general comments:

In conclusion, lens status may affect the tube position but, in our study, phakia/pseudophakia numbers and anterior chamber depths were not significantly different between two groups, and majority of eyes were pseudophakia at the time of tube insertion.

ECD loss after cataract surgery is evident but the period of significant decline varies from study to study, although most study claimed that ECD loss was approached at physiological loss level after three months. Thus, an additional limitation existed in our study because lens status and numbers of combined cataract surgery between two groups were not the same. However, we tried to minimize this limitation by strictly obeying the exclusion criteria, and there was no significant difference in the number of triple surgeries between two groups (p = 0.077, Table 1). 

We added sentences in Discussion to include points above: line 286-287. Line 348-353.

9. Results – all tables and figures should have figures presented with only 1 decimal point.

Thank you for pointing out.

We looked up other published papers in this journal and The PLoS ONE guideline for Table section showing an example. (table below) Accordingly, figures in Table 1,2,3, and 4 were revised. 

In addition, TL, TCD, and TCA in Figure 3C and 3D could not be revised because these figures were measured and displayed using a built-in ruler and angle indicator, as mentioned in Methods. 

10. Table 2 – what is the mean follow us in each group? This could also affect ECD count.

The mean follow-up period is shown in Table 1: 15.84 ± 6.49 months for the gAGV group and 18.39 ± 5.41 months for the ngAGV group (p = 0.100). There is no significant difference between two groups. 

11. Discussion – points for discussion should added as suggested above.

Added and revised sentences were indicated in the answers to each question.

Reviewer #2: The authors claim that the postoperative loss of corneal endothelial cells was less in the case of guiding the tube placement during Ahmed valve implantation. Overall, I think it's an interesting study. However, since this is a retrospective study, further analysis and description would be needed. In the case of guiding a tube, it seems necessary to explain why the guiding was conducted. It should be described whether the surgical method changed by period or only in what cases. In addition, it would be better to further analyze the power of analysis according to the number of subjects.

Thank you for your encouragement and comments. 

This is a retrospective study and we tried to discuss limitations thoroughly in this paper. Differences in the group size and follow-up periods between two groups were mainly due to a gradual shift in surgical methods. Nevertheless, the group size was sufficient in this comparative analysis, and statistically significance was well proven in results. Previous study comparing anterior chamber vs ciliary sulcus AGV tube placement also had different group sizes due to gradual change in surgical methods (ref. 27 in manuscript). In early 2020, the surgeon changed the method from non-guided to guided AGV implantation after pilot study comparing ECD loss between two groups, and two surgical methods were not used together during the same period. The sentence in Methods line 121-122 was revised to make this point clearer:

The method was gradually changed to include the guided approach since 2020. �

The method was gradually changed from non-guided to guided AGV implantation since 2020 and two surgical methods were not used together during the same period.

There was a period of learning curve with the combined use of two surgical methods, and cases during this period (one month) were excluded as described in Discussion line 347-348. It is believed to be the best way to randomize a single surgeon’s surgical methods since patients were assigned to guided or non-guided groups depending on the specific period they had surgery, not on their age, sex, glaucoma diagnosis or glaucoma severity. Thus, there was no selection bias between the two groups, as mentioned in Discussion line 345-347.

The 4-0 nylon stent plays a very important role as a guide in this technique. As mentioned in the introduction, it supports the flexible tube from kinking and bending when the tube enters the AC. Even if we identify the ideal entry site in the AC with a spatula and create the scleral tunnel with a bent 23-G needle, we struggle much to insert the tube along the scleral tunnel because the tube is easily bent, unlike the firm needle. Moreover, when we try to insert the tube close to the peripheral iris, the tube is easily stuck by iris increasing the risk of iridodialysis, or wrong insertion to the sulcus. Therefore, it is beneficial to use a 4-0 nylon stent to firmly support the tube from the inside to make sure the tube is inserted at the ideal position. This was also reflected in surgical time as written in Discussion line 295-300; the ngAGV group showed a greater deviation in the mean surgical time, and in most of these cases, the tube was kinked and bent and suffered a lot of time during insertion to the AC. 

It should be also noted that the prevalence of PACG is higher in Asians than others6, and ethical differences show Asians have narrower and more crowded AC structures that other races7 in previous studies. Therefore, using a guide to support the tube during insertion may have more benefits in Asians as in our study, or in patients having narrow AC structure. 

As you pointed out, further studies are needed to find out long-term results beyond 2 years (in Discussion line 359-360). More number of eyes in each group should be included in further analysis, and it may enable more accurate comparative analysis. 

6Primary angle closure glaucoma: What we know and what we don't know. Sun X, Dai Y, Chen Y, Yu DY, Cringle SJ, Chen J, et al. Prog Retin Eye Res. 2017;57:26-45.

7Ethnic difference of the anterior chamber area and volume and its association with angle width. Wang D, Qi M, He M, Wu L, Lin S. Invest Ophthalmol Vis Sci. 2012;53:3139-44.

Finally, we hope that our study will show ophthalmologists the appropriate method to reduce the rate of ECD loss after AGV surgery and prevent corneal decompensation in future patients.

Chang Kyu Lee, MD, PhD

Associate professor

Department of Ophthalmology, glaucoma specialty

Ulsan University Hospital, University of Ulsan College of Medicine,

Ulsan, South Korea

---

## [Decision Letter · Decision Letter 1]

15 Nov 2022

Effect of Guided Ahmed Glaucoma Valve Implantation on Corneal Endothelial Cells: A 2-Year Comparative Study

PONE-D-22-22914R1

Dear Dr. Lee,

We’re pleased to inform you that your manuscript has been judged scientifically suitable for publication and will be formally accepted for publication once it meets all outstanding technical requirements.

Kind regards,

Asaf Achiron

Academic Editor

PLOS ONE

Additional Editor Comments (optional):

Reviewers' comments:

Reviewer's Responses to Questions

**Comments to the Author**

1. If the authors have adequately addressed your comments raised in a previous round of review and you feel that this manuscript is now acceptable for publication, you may indicate that here to bypass the “Comments to the Author” section, enter your conflict of interest statement in the “Confidential to Editor” section, and submit your "Accept" recommendation.

Reviewer #1: All comments have been addressed

Reviewer #2: All comments have been addressed

2. Is the manuscript technically sound, and do the data support the conclusions?

Reviewer #1: Yes

Reviewer #2: Yes

3. Has the statistical analysis been performed appropriately and rigorously? 

Reviewer #1: Yes

Reviewer #2: Yes

4. Have the authors made all data underlying the findings in their manuscript fully available?

Reviewer #1: Yes

Reviewer #2: Yes

5. Is the manuscript presented in an intelligible fashion and written in standard English?

Reviewer #1: Yes

Reviewer #2: Yes

6. Review Comments to the Author

Reviewer #1: The replies to the comments presented were reasonable. Even though this retrospective study has major limitations, they were mentioned in the revised manuscript.

Reviewer #2: The authors has adequately addressed everything pointed out previously. No more concerns about publication.

7. PLOS authors have the option to publish the peer review history of their article (what does this mean?). If published, this will include your full peer review and any attached files.

Reviewer #1: No

Reviewer #2: No

---

## [Editor Report · Acceptance letter]

22 Nov 2022

PONE-D-22-22914R1 

Effect of Guided Ahmed Glaucoma Valve Implantation
on Corneal Endothelial Cells: A 2-Year Comparative Study 

Dear Dr. Lee:

I'm pleased to inform you that your manuscript has been deemed suitable for publication in PLOS ONE. Congratulations! Your manuscript is now with our production department. 

Kind regards, 

on behalf of

Dr. Asaf Achiron 

Academic Editor

PLOS ONE